# Quantum theory cannot consistently describe the use of itself

Daniela Frauchiger[1] & Renato Renner[1]

Quantum theory provides an extremely accurate description of fundamental processes in physics. It thus seems likely that the theory is applicable beyond the, mostly microscopic, domain in which it has been tested experimentally. Here, we propose a Gedankenexperiment to investigate the question whether quantum theory can, in principle, have universal validity. The idea is that, if the answer was yes, it must be possible to employ quantum theory to model complex systems that include agents who are themselves using quantum theory. Analysing the experiment under this presumption, we find that one agent, upon observing a particular measurement outcome, must conclude that another agent has predicted the opposite outcome with certainty. The agents' conclusions, although all derived within quantum theory, are thus inconsistent. This indicates that quantum theory cannot be extrapolated to complex systems, at least not in a straightforward manner.

---

[1] Institute for Theoretical Physics, ETH Zurich, 8093 Zurich, Switzerland. Correspondence and requests for materials should be addressed to R.R. (email: renner@ethz.ch)

**D**irect experimental tests of quantum theory are mostly restricted to microscopic domains. Nevertheless, quantum theory is commonly regarded as being (almost) universally valid. It is not only used to describe fundamental processes in particle and solid state physics, but also, for instance, to explain the cosmic microwave background or the radiation of black holes.

The presumption that the validity of quantum theory extends to larger scales has remarkable consequences, as noted already in 1935 by Schrödinger[1]. His famous example consisted of a cat that is brought into a state corresponding to a superposition of two macroscopically entirely different states, one in which it is dead and one in which it is alive. Schrödinger pointed out, however, that such macroscopic superposition states do not represent anything contradictory in themselves.

This view was not shared by everyone. In 1967, Wigner proposed an argument, known as the Wigner's Friend Paradox, which should show that "quantum mechanics cannot have unlimited validity"[2]. His idea was to consider the views of two different observers in an experiment analogous to the one depicted in Fig. 1. One observer, called agent F, measures the vertical polarisation $z$ of a spin one-half particle S, such as a silver atom. Upon observing the outcome, which is either $z = -\frac{1}{2}$ or $z = +\frac{1}{2}$, agent F would thus say that S is in state

$$\psi_S = |\downarrow\rangle_S \quad \text{or} \quad \psi_S = |\uparrow\rangle_S, \tag{1}$$

respectively. The other observer, agent W, has no direct access to the outcome $z$ observed by his friend F. Agent W could instead model agent F's lab as a big quantum system, $L \equiv S \otimes D \otimes F$, which contains the spin S as a subsystem, another subsystem, D, for the friend's measurement devices and everything else connected to them, as well as a subsystem F that includes the friend herself. Suppose that, from agent W's perspective, the lab L is initially in a pure state and that it remains isolated during agent F's spin measurement experiment. (One may object that these assumptions are unrealistic[3], but, crucially, the laws of quantum theory do not preclude that they be satisfied to arbitrarily good approximation[4].) Translated to quantum mechanics, this means that the dependence of the final state of L on the initial state of S is described by a linear map of the form

$$U_{S \to L} = \begin{cases} \{|\downarrow\rangle_S \mapsto |-\tfrac{1}{2}\rangle_L \equiv |\downarrow\rangle_S \otimes |\text{``}z = -\tfrac{1}{2}\text{''}\rangle_D \otimes |\text{``}\psi_S = |\downarrow\rangle\text{''}\rangle_F \\ \{|\uparrow\rangle_S \mapsto |+\tfrac{1}{2}\rangle_L \equiv |\uparrow\rangle_S \otimes |\text{``}z = +\tfrac{1}{2}\text{''}\rangle_D \otimes |\text{``}\psi_S = |\uparrow\rangle\text{''}\rangle_F \end{cases}. \tag{2}$$

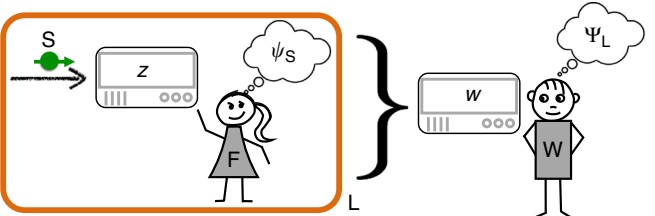

**Fig. 1** Wigner's and Deutsch's arguments. Agent F measures the spin S of a silver atom in the vertical direction, obtaining outcome $z$. From F's perspective, S is then in one of the two pure states $\psi_S$ given in (1). Agent W, who is outside of F's lab, may instead regard that lab, including the agent F, as a big quantum system L (orange box). Wigner argued that, having no access to $z$, he would assign a superposition state $\Psi_L$ of the form (3) to L[2]. Deutsch later noted that agent W could in principle test this state assignment by applying a carefully designed measurement to L[6]

Here, $|\text{``}z = -\tfrac{1}{2}\text{''}\rangle_D$ and $|\text{``}z = +\tfrac{1}{2}\text{''}\rangle_D$ denote states of D depending on the measurement outcome $z$ shown by the devices within the lab. Analogously, $|\text{``}\psi_S = |\downarrow\rangle\text{''}\rangle_F$ and $|\text{``}\psi_S = |\uparrow\rangle\text{''}\rangle_F$ are states of F, which we may label by the friend's own knowledge of $\psi_S$; cf. (1). Now, suppose agent W knew that the spin was initialized to $|\rightarrow\rangle_S \equiv \sqrt{1/2}(|\downarrow\rangle_S + |\uparrow\rangle_S)$ before agent F measured it. Then, by linearity, the final state that agent W would assign to L is

$$\Psi_L = \sqrt{\tfrac{1}{2}}\left(|-\tfrac{1}{2}\rangle_L + |+\tfrac{1}{2}\rangle_L\right), \tag{3}$$

i.e., a linear superposition of the two macroscopically distinct states defined in (2). To compare this to agent F's view (1), one must consider the restriction of (3) to S. The latter is a maximally mixed state, and thus obviously different from agent F's pure state assignment (1). But, crucially, the difference can be explained by the two agents' distinct level of knowledge: Agent F has observed $z$ and hence knows the final spin direction, whereas agent W is ignorant about it[5]. Consequently, although the superposition state (3) may appear "absurd"[2], it does not contradict (1). For this reason, the Wigner's Friend Paradox cannot be regarded as an argument that rules out quantum mechanics as a universally valid theory.

In this work we propose a Gedankenexperiment that extends Wigner's setup. It consists of agents who are using quantum theory to reason about other agents who are also using quantum theory. Our main finding is that such a self-referential use of the theory yields contradictory claims. This result can be phrased as a no-go theorem (Theorem 1). It asserts that three natural-sounding assumptions, (Q), (C), and (S), cannot all be valid. Assumption (Q) captures the universal validity of quantum theory (or, more specifically, that an agent can be certain that a given proposition holds whenever the quantum-mechanical Born rule assigns probability-1 to it). Assumption (C) demands consistency, in the sense that the different agents' predictions are not contradictory. Finally, (S) is the requirement that, from the viewpoint of an agent who carries out a particular measurement, this measurement has one single outcome. The theorem itself is neutral in the sense that it does not tell us which of these three assumptions is wrong. However, it implies that any specific interpretation of quantum theory, when applied to the Gedankenexperiment, will necessarily conflict with at least one of them. This gives a way to test and categorise interpretations of quantum theory.

## Results

**The Gedankenexperiment.** In the setup considered by Wigner (cf. Fig. 1), agent F carries out her measurement of S in a perfectly isolated lab L, so that the outcome $z$ remains unknown to anyone else. The basic idea underlying the Gedankenexperiment we present here is to make some of the information about $z$ available to the outside—but without lifting the isolation of L. Roughly, this is achieved by letting the initial state of S depend on a random value, $r$, which is known to another agent outside of L.

Box 1 specifies the proposed Gedankenexperiment as a stepwise procedure. The steps are to be executed by different agents—four in total. Two of them, the "friends" F and $\overline{F}$, are located in separate labs, denoted by L and $\overline{L}$, respectively. The two other agents, W and $\overline{W}$, are at the outside, from where they can apply measurements to L and $\overline{L}$, as shown in Fig. 2. We assume that L and $\overline{L}$ are, from the viewpoint of the agents W and $\overline{W}$, initially in a pure state, and that they remain isolated during the experiment unless the protocol explicitly prescribes a communication step or a measurement applied to them. Note that the experiment can be described within standard quantum-mechanical formalism, with each step corresponding to a fixed evolution map acting on particular subsystems (cf. the circuit diagram in the Methods section).

**Box 1: Experimental procedure**

The steps are repeated in rounds $n = 0, 1, 2, \ldots$ until the halting condition in the last step is satisfied. The numbers on the left indicate the timing of the steps, and we assume that each step takes at most one unit of time. (For example, in round $n = 0$, agent F starts her measurement of S at time 0:10 and completes it before time 0:11.) Definitions of the relevant state and measurement basis vectors are provided in Tables 1 and 2.

At $n$:00    Agent $\overline{F}$ invokes a randomness generator (based on the measurement of a quantum system R in state $|init\rangle_R$ as defined in Table 1) that outputs $r = $ heads or $r = $ tails with probabilities $\frac{1}{3}$ and $\frac{2}{3}$, respectively. She sets the spin S of a particle to $|\downarrow\rangle_S$ if $r = $ heads and to $|\rightarrow\rangle_S \equiv \sqrt{1/2}\,(|\downarrow\rangle_S + |\uparrow\rangle_S)$ if $r = $ tails, and sends it to F.

At $n$:10    Agent F measures S w.r.t. the basis $\{|\downarrow\rangle_S, |\uparrow\rangle_S\}$, recording the outcome $z \in \{-\frac{1}{2}, +\frac{1}{2}\}$.

At $n$:20    Agent $\overline{W}$ measures lab $\overline{L}$ w.r.t. a basis containing the vector $|ok\rangle_{\overline{L}}$ (defined in Table 2). If the outcome associated to this vector occurs he announces $\overline{w} = \overline{ok}$ and else $\overline{w} = \overline{fail}$.

At $n$:30    Agent W measures lab L w.r.t. a basis containing the vector $|ok\rangle_L$ (defined in Table 2). If the outcome associated to this vector occurs he announces $w = ok$ and else $w = fail$.

At $n$:40    If $\overline{w} = \overline{ok}$ and $w = ok$ then the experiment is halted.

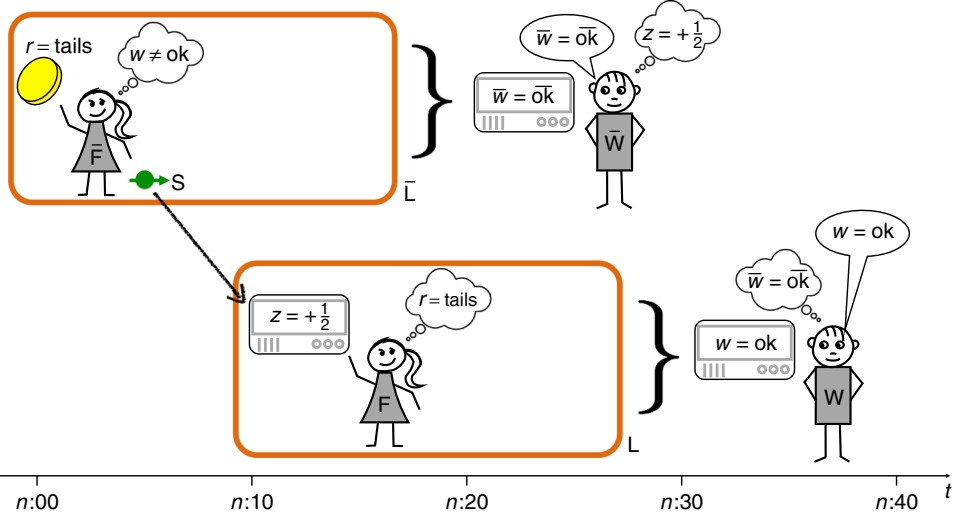

**Fig. 2** Illustration of the Gedankenexperiment. In each round $n = 0, 1, 2, \ldots$ of the experiment, agent $\overline{F}$ tosses a coin and, depending on the outcome $r$, polarises a spin particle S in a particular direction. Agent F then measures the vertical polarisation $z$ of S. Later, agents $\overline{W}$ and W measure the entire labs $\overline{L}$ and L (where the latter includes S) to obtain outcomes $\overline{w}$ and $w$, respectively. For the analysis of the experiment, we assume that all agents are aware of the entire procedure as specified in Box 1, but they are located at different places and therefore make different observations. Agent F, for instance, observes $z$ but has no direct access to $r$. She may however use quantum theory to draw conclusions about $r$

As indicated by the term Gedankenexperiment, we do not claim that the experiment is technologically feasible, at least not in the form presented here. Like other thought experiments, its purpose is not to probe nature, but rather to scrutinise the consistency of our currently best available theories that describe nature—in this case quantum theory. (One may compare this to, say, the Gedankenexperiment of letting an observer cross the event horizon of a black hole. Although we do not have the technology to carry out this experiment, reasoning about it provides us with insights on relativity theory.)

Before proceeding to the analysis of the experiment, a few comments about its relation to earlier proposals are in order. In the case where $r = $ tails, agent F receives S prepared in state $|\rightarrow\rangle_S$. The first part of the experiment, prior to the measurements carried out by the agents $\overline{W}$ and W, is then equivalent to Wigner's original experiment as described in the section Introduction[2]. Furthermore, adding to this the measurement of agent F's lab by agent W, one retrieves an extension of Wigner's experiment proposed by Deutsch[6] (Fig. 1). The particular procedure of how agent F prepares the spin S in the first step described in Box 1, as well as the choice of measurements, is motivated by a construction due to Hardy[7,8], known as Hardy's Paradox. The

setup considered here is also similar to a proposal by Brukner[9], who used a modification of Wigner's argument to obtain a strengthening of Bell's theorem[10] (cf. Discussion section).

**Analysis of the Gedankenexperiment.** We analyse the experiment from the viewpoints of the four agents, $\overline{F}$, F, $\overline{W}$, and W, who have access to different pieces of information (cf. Fig. 2). We assume, however, that all agents are aware of the entire experimental procedure as described in Box 1, and that they all employ the same theory. One may thus think of the agents as computers that, in addition to carrying out the steps of Box 1, are programmed to draw conclusions according to a given set of rules. In the following, we specify these rules as assumptions (Boxes 2–4).

The first such assumption, Assumption (Q) is that any agent A "uses quantum theory." By this we mean that A may predict the outcome of a measurement on any system S around him via the quantum-mechanical Born rule. For our purposes, it suffices to consider the special case where the state $|\psi\rangle_S$ that A assigns to S lies in the image of only one of the measurement operators $\pi_x^{t_0}$, say the one with $x = \xi$. In this case, the Born rule asserts that the outcome $x$ equals $\xi$ with certainty; see Box 2.

Crucially, S may be a large and complex system, even one that itself contains agents. In fact, to start our analysis, we take the

---

**Box 2: Assumption (Q)**

Suppose that agent A has established that

*Statement* $A^{(i)}$: "System S is in state $|\psi\rangle_S$ at time $t_0$."

Suppose furthermore that agent A knows that

*Statement* $A^{(ii)}$: "The value $x$ is obtained by a measurement of S w.r.t. the family $\{\pi_x^{t_0}\}_{x \in \mathcal{X}}$ of Heisenberg operators relative to time $t_0$, which is completed at time $t$."

If $\langle\psi|\pi_\xi^{t_0}|\psi\rangle = 1$ for some $\xi \in \mathcal{X}$ then agent A can conclude that

*Statement* $A^{(iii)}$: "I am certain that $x = \xi$ at time $t$."

---

system S to be the entire lab L, which in any round $n$ of the experiment is measured with respect to the Heisenberg operators $\pi_{w=\text{ok}}^{n:10}$ and $\pi_{w=\text{fail}}^{n:10}$ defined in Table 2. Suppose that agent $\overline{F}$ wants to predict the outcome $w$ of this measurement. To this aim, she may start her reasoning with a statement that describes the corresponding measurement.

*Statement* $\overline{F}^{n:00}$: "The value $w$ is obtained by a measurement of L w.r.t. $\{\pi_{w=\text{ok}}^{n:10}, \pi_{w=\text{fail}}^{n:10}\}$, which is completed at time $n{:}31$."

Here and in the following, we specify for each statement a time, denoted as a superscript, indicating when the agent could have inferred the statement. Agent $\overline{F}$'s statement $\overline{F}^{n:00}$ above does not depend on any observations, so the time $n{:}00$ we have assigned to it is rather arbitrary. This is, however, different for the next statement, which is based on knowledge of the value $r$. Suppose that agent $\overline{F}$ got $r =$ tails as the output of the random number generator in round $n$. According to the experimental instructions, she will then prepare the spin S in state $|\rightarrow\rangle_S$. Now, after completing the preparation, say at time $n{:}01$, she may make a second statement, taking into account that S remains unchanged until F starts her measurement at time $n{:}10$.

*Statement* $\overline{F}^{n:01}$: "The spin S is in state $|\rightarrow\rangle_S$ at time $n{:}10$."

Agent $\overline{F}$ could conclude from this that the later state of the lab L, $U_{S\rightarrow L}^{10\rightarrow 20}|\rightarrow\rangle_S = \sqrt{\frac{1}{2}}\left(|-\frac{1}{2}\rangle_L + |+\frac{1}{2}\rangle_L\right)$, will be orthogonal to $|\text{ok}\rangle_L$. An equivalent way to express this is that the state $|\rightarrow\rangle_S$ has no overlap with the Heisenberg measurement operator corresponding to outcome $w = \text{ok}$, i.e.,

$$\langle\rightarrow|\pi_{w=\text{fail}}^{n:10}|\rightarrow\rangle = 1 - \langle\rightarrow|\pi_{w=\text{ok}}^{n:10}|\rightarrow\rangle = 1 . \quad (4)$$

The two statements $\overline{F}^{n:00}$ and $\overline{F}^{n:01}$, inserted into (Q), thus imply that $w =$ fail. We may assume that agent $\overline{F}$ draws this conclusion at time $n{:}02$ and, for later use, put it down as statement $\overline{F}^{n:02}$ in Table 3. Similarly, agent F's reasoning may be based upon a description of her spin measurement, which is defined by the operators $\pi_{z=-\frac{1}{2}}^{n:10}$ and $\pi_{z=+\frac{1}{2}}^{n:10}$ given in Table 2.

*Statement* $F^{n:10}$: "The value $z$ is obtained by a measurement of the spin S w.r.t. $\{\pi_{z=-\frac{1}{2}}^{n:10}, \pi_{z=+\frac{1}{2}}^{n:10}\}$, which is completed at time $n{:}11$."

Suppose now that agent F observed $z = +\frac{1}{2}$ in round $n$. Since, by definition,

$$\langle\downarrow|\pi_{z=-\frac{1}{2}}^{n:10}|\downarrow\rangle = 1 \quad (5)$$

it follows from (Q) that S was not in state $|\downarrow\rangle$, and hence that the random value $r$ was not heads. This is statement $F^{n:12}$ of Table 3. We proceed with agent $\overline{W}$, who may base his reasoning upon his knowledge of how the random number generator was initialised.

*Statement* $\overline{W}^{n:21}$: "System R is in state $|\text{init}\rangle_R$ at time $n{:}00$."

Consider the event that $\overline{w} = \overline{\text{ok}}$ and $z = -\frac{1}{2}$, as well as its complement. The Heisenberg operators of the corresponding measurement are given in Table 2. It is straightforward to verify that $U_{R\rightarrow LS}^{00\rightarrow 10}|\text{init}\rangle_R = \sqrt{\frac{1}{3}}|\overline{\text{h}}\rangle_{\overline{L}}\otimes|\downarrow\rangle_S + \sqrt{\frac{2}{3}}|\overline{\text{t}}\rangle_{\overline{L}}\otimes|\rightarrow\rangle_S$ is orthogonal to $|\overline{\text{ok}}\rangle_{\overline{L}}\otimes|\downarrow\rangle_S$, which implies that

$$\langle\text{init}|\pi_{(\overline{w},z)\neq(\overline{\text{ok}},-\frac{1}{2})}^{n:00}|\text{init}\rangle = 1 - \langle\text{init}|\pi_{(\overline{w},z)=(\overline{\text{ok}},-\frac{1}{2})}^{n:00}|\text{init}\rangle = 1 . \quad (6)$$

Agent $\overline{W}$, who also uses (Q), can hence be certain that $(\overline{w},z)\neq(\overline{\text{ok}}, -\frac{1}{2})$. This implies that statement $\overline{W}^{n:22}$ of Table 3 holds whenever $\overline{w} = \overline{\text{ok}}$. Furthermore, because agent $\overline{W}$ announces $\overline{w}$, agent W can be certain about $\overline{W}$'s knowledge, which justifies statement $W^{n:26}$ of the table. We have thus established all statements in the third column of Table 3.

For later use we also note that a simple calculation yields

$$\langle\text{init}|\pi_{(\overline{w},w)=(\overline{\text{ok}},\text{ok})}^{n:00}|\text{init}\rangle = \frac{1}{12} \quad (7)$$

where $\pi_{(\overline{w},w)=(\overline{\text{ok}},\text{ok})}^{n:00}$ is the Heisenberg operator belonging to the event that $\overline{w} = \overline{\text{ok}}$ and $w = \text{ok}$, as defined in Table 2. Hence, according to quantum mechanics, agent W can be certain that the outcome $(\overline{w},w) = (\overline{\text{ok}},\text{ok})$ occurs after finitely many rounds. This corresponds to the following statement (which can indeed be derived using (Q), as shown in the Methods section).

*Statement* $W^{0:00}$: "I am certain that there exists a round $n$ in which the halting condition at time $n{:}40$ is satisfied."

The agents may now obtain further statements by reasoning about how they would reason from the viewpoint of other agents, as illustrated in Fig. 3. To enable such nested reasoning we need another assumption, Assumption (C); see Box 3.

Agent F may insert agent $\overline{F}$'s statement $\overline{F}^{n:02}$ into $F^{n:12}$, obtaining statement $F^{n:13}$ in Table 3. By virtue of (C), she may then conclude that statement $F^{n:14}$ holds, too. Similarly, $\overline{W}$ may combine this latter statement with his statement $\overline{W}^{n:22}$ to obtain $\overline{W}^{n:23}$. He could then, again using (C), conclude that statement $\overline{W}^{n:24}$ holds. Finally, agent W can insert this into his statement $W^{n:26}$ to obtain statement $W^{n:27}$ and, again with (C), statement $W^{n:28}$. This completes the derivation of all statements in Table 3.

For the last part of our analysis, we take again agent W's perspective. According to statement $W^{n:00}$, the experiment has a final round $n$ in which the halting condition will be satisfied, meaning in particular that agent $\overline{W}$ announces $\overline{w} = \overline{\text{ok}}$. Agent W infers from this that statement $W^{n:28}$ of Table 3 holds in that round, i.e., he is certain that he will observe $w =$ fail at time $n{:}31$. However, in this final round, he will nevertheless observe $w = \text{ok}$! We have thus reached a contradiction—unless agent W would

accept that $w$ simultaneously admits multiple values. For our discussion below, it will be useful to introduce an explicit assumption, termed Assumption (S), which disallows this; see Box 4.

**No-go theorem.** The conclusion of the above analysis may be phrased as a no-go theorem.

*Theorem 1.* Any theory that satisfies assumptions (Q), (C), and (S) yields contradictory statements when applied to the Gedankenexperiment of Box 1.

To illustrate the theorem, we consider in the following different interpretations and modifications of quantum theory. Theorem 1 implies that any of them must violate either (Q), (C), or (S). This yields a natural categorisation as shown in Table 4 and discussed in the following subsections.

**Theories that violate Assumption (Q).** Assumption (Q) corresponds to the quantum-mechanical Born rule. Since the assumption is concerned with the special case of probability-1 predictions only, it is largely independent of interpretational questions, such as the meaning of probabilities in general. However, the nontrivial aspect of (Q) is that it regards the Born rule as a universal law. That is, it demands that an agent A can apply the rule to arbitrary systems S around her, including large ones that may contain other agents. The specifier "around" is crucial, though: Assumption (Q) does not demand that agent A

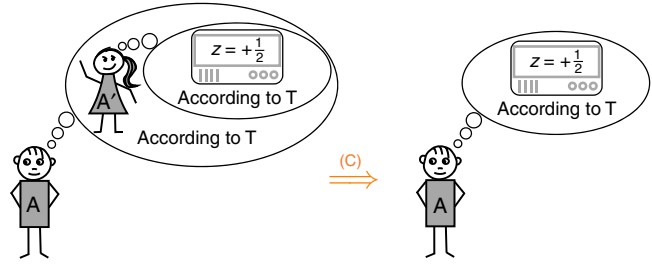

**Fig. 3** Consistent reasoning as required by Assumption (C). If a theory T (such as quantum theory) enables consistent reasoning (C) then it must allow any agent A to promote the conclusions drawn by another agent A' to his own conclusions, provided that A' has the same initial knowledge about the experiment and reasons within the same theory T. A classical example of such recursive reasoning is the muddy children puzzle (here T is just standard logic; see ref. 11 for a detailed account). The idea of using a physical theory T to describe agents who themselves use T has also appeared in thermodynamics, notably in discussions around Maxwell's demon[12]

can describe herself as a quantum system. Such a requirement would indeed be overly restrictive (see ref. 13) for it would immediately rule out interpretations in the spirit of Copenhagen, according to which the observed quantum system and the observer must be distinct from each other[14,15].

Assumption (Q) is manifestly violated by theories that postulate a modification of standard quantum mechanics, such as spontaneous[16–20] and gravity-induced[21–23] collapse models (cf. 24 for a review). These deviate from the standard theory already on microscopic scales, although the effects of the deviation typically only become noticeable in larger systems.

In some approaches to quantum mechanics, it is simply postulated that large systems are "classical", but the physical mechanism that explains the absence of quantum features remains unspecified[25]. In the view described in ref. 3, for instance, the postulate says that measurement devices are infinite-dimensional systems whereas observables are finite. This ensures that coherent and incoherent superpositions in the state of a measurement device are indistinguishable. Similarly, according to the "ETH approach"[26], the algebra of available observables is time-dependent and does not allow one to distinguish coherent from incoherent superpositions once a measurement has been completed. General measurements on systems that count themselves as measurement devices are thus ruled out. Another example is the "CSM ontology"[27], according to which measurements must always be carried out in a "context", which includes the measurement devices. It is then postulated that this context cannot itself be treated as a quantum system. Within all these interpretations, the Born rule still holds "for all practical purposes", but is no longer a universally applicable law in the sense of Assumption (Q) (see the discussion in ref. 4).

Another class of theories that violate (Q), although in a less obvious manner, are particular "hidden-variable (HV) interpretations"[28], with "Bohmian mechanics" as the most prominent example[29–31]. According to the common understanding, Bohmian mechanics is a "theory of the universe" rather than a theory about subsystems[32]. This means that agents who apply the theory must in principle always take an outside perspective on the entire universe, describing themselves as part of it. This outside perspective is identical for all agents, which ensures consistency and hence the validity of Assumption (C). However, because (S) is satisfied, too, it follows from Theorem 1 that (Q) must be violated (see the Methods section for more details).

**Theories that violate Assumption (C).** If a theory satisfies (Q) and (S) then, by Theorem 1, it must violate (C). This conclusion applies to a wide range of common readings of quantum mechanics, including most variants of the Copenhagen

---

**Box 3: Assumption (C)**

Suppose that agent A has established that
 *Statement* A^(i): "I am certain that agent A', upon reasoning within the same theory as the one I am using, is certain that $x = \xi$ at time $t$."
Then agent A can conclude that
 *Statement* A^(ii): "I am certain that $x = \xi$ at time $t$."

---

**Box 4: Assumption (S)**

Suppose that agent A has established that
 *Statement* A^(i): "I am certain that $x = \xi$ at time $t$."
Then agent A must necessarily deny that
 *Statement* A^(ii): "I am certain that $x \neq \xi$ at time $t$."

---

interpretation. One concrete example is the "consistent histories" (CH) formalism[33–35], which is also similar to the "decoherent histories" approach[36,37]. Another class of examples are subjectivistic interpretations, which regard statements about outcomes of measurements as personal to an agent, such as "relational quantum mechanics"[38], "QBism"[39,40], or the approach proposed in ref. [9] (see Methods section for a discussion of the CH formalism as well as QBism).

The same conclusion applies to HV interpretations of quantum mechanics, provided that we use them to describe systems around us rather than the universe as a whole (contrasting the paradigm of Bohmian mechanics discussed above). In this case, both (Q) and (S) hold by construction. This adds another item to the long list of no-go results for HV interpretations: they cannot be local[10], they must be contextual[41,42], and they violate freedom of choice[43,44]. Theorem 1 entails that they also violate (C). In particular, there cannot exist an assignment of values to the HVs that is consistent with the agents' conclusions.

**Theories that violate Assumption (S).** Although intuitive, (S) is not implied by the bare mathematical formalism of quantum mechanics. Among the theories that abandon the assumption are the "relative state formulation" and "many-worlds interpretations"[6,45–48]. According to the latter, any quantum measurement results in a branching into different "worlds", in each of which one of the possible measurement outcomes occurs. Further developments and variations include the "many-minds interpretation"[49,50] and the "parallel lives theory"[51]. A related concept is "quantum Darwinism"[52], whose purpose is to explain the perception of classical measurement outcomes in a unitarily evolving universe.

While many-worlds interpretations manifestly violate (S), their compatibility with (Q) and (C) depends on how one defines the branching. If one regards it as an objective process, (Q) may be violated (cf. the example in Section 10 of ref. [53]). It is also questionable whether (Q) can be upheld if branches do not persist over time (cf. the no-histories view described in ref. [54]).

**Implicit assumptions.** Any no-go result, as for example Bell's theorem[10], is phrased within a particular framework that comes with a set of built-in assumptions. Hence it is always possible that a theory evades the conclusions of the no-go result by not fulfilling these implicit assumptions. Here we briefly discuss how Theorem 1 compares in this respect to other results in the literature.

Bell's original work[10] treats probabilities as a primitive notion. Similarly, many of the modern arguments in quantum foundations employ probabilistic frameworks[55–62]. In contrast, probabilities are not used in the argument presented here—although Assumption (Q) is of course motivated by the idea that a statement can be regarded as "certain" if the Born rule assigns probability-1 to it. In particular, Theorem 1 does not depend on how probabilities different from 1 are interpreted.

Another distinction is that the framework used here treats all statements about observations as subjective, i.e., they are always defined relative to an agent. This avoids the a priori assumption that measurement outcomes obtained by different agents simultaneously have definite values. (Consider for example Wigner's original setup described in section Introduction. Even when Assumptions (C) and (S) hold, agent W is not forced to assign a definite value to the outcome $z$ observed by agent F.) The assumption of simultaneous definiteness is otherwise rather common. It not only enters the proof of Bell's theorem[10] but also the aforementioned arguments based on probabilistic frameworks.

Nevertheless, in our considerations, we used concepts such as that of an "agent" or of "time". It is conceivable that the conclusions of Theorem 1 can be avoided by theories that provide a nonstandard understanding of these concepts. We are, however, not aware of any concrete examples of such theories.

## Discussion

In the Gedankenexperiment proposed in this article, multiple agents have access to different pieces of information, and draw conclusions by reasoning about the information held by others. In the general context of quantum theory, the rules for such nested reasoning may be ambiguous, for the information held by one agent can, from the viewpoint of another agent, be in a superposition of different "classical" states. Crucially, however, in the argument presented here, the agents' conclusions are all restricted to supposedly unproblematic "classical" cases. For example, agent $\overline{W}$ only needs to derive a statement about agent F in the case where, conditioned on his own information $\bar{w}$, the information $z$ held by F has a well-defined value (Table 3). Nevertheless, as we have shown, the agents arrive at contradictory statements.

Current interpretations of quantum theory do not agree on the origin of this contradiction (cf. Table 4). To compare the different views, it may therefore be useful to rephrase the experiment as a concrete game-theoretic decision problem. Suppose that a casino offers the following gambling game. One round of the experiment of Box 1 is played, with the gambler in the role of agent W, and the roles of $\overline{F}$, F, and $\overline{W}$ taken by employees of the casino. The casino promises to pay €1000 to the gambler if F's random value was $r =$ heads. Conversely, if $r =$ tails, the gambler must pay €500 to the casino. It could now happen that, at the end of the game, $w =$ ok and $\overline{w} = \overline{ok}$, and that a judge can convince herself of this outcome. The gambler and the casino are then likely to end up in a dispute, putting forward arguments taken from Table 3.

*Gambler*: "The outcome $w =$ ok proves, due to (4), that S was not prepared in state $|\rightarrow\rangle_S$. This means that $r =$ heads and hence the casino must pay me €1000."

*Casino*: "The outcome $\overline{w} = \overline{ok}$ implies, due to (6), that our employee observed $z = +\frac{1}{2}$. This in turn proves that S was not prepared in state $|\downarrow\rangle_S$. But this means that $r =$ tails, so the gambler must pay us €500."

How should the judge decide on this case? Could it even be that both assertions must be accepted as two "alternative facts" about what the value $r$ was? We leave it as a task for further research to explore what the different interpretations of quantum mechanics have to say about this game.

Theorem 1 may be compared to earlier no-go results, such as[7–10,41–43], which also use assumptions similar to (Q) and (S) (although the latter is often implicit). These two assumptions are usually shown to be in conflict with additional assumptions about reality, locality, or freedom of choice. For example, the result of ref. [9], which is as well based on an extension of Wigner's argument, asserts that no theory can fulfil all of the following properties: (i) be compatible with quantum theory on all scales, (ii) simultaneously assign definite truth values to measurement outcomes of all agents, (iii) allow agents to freely choose measurement settings, and (iv) be local. Here, we have shown that Assumptions (Q) and (S) are already problematic by themselves, in the sense that agents who use these assumptions to reason about each other as in Fig. 3 will arrive at inconsistent conclusions.

Another noticeable difference to earlier no-go results is that the argument presented here does not employ counterfactual reasoning. That is, it does not refer to choices that could have been made but have not actually been made. In fact, in the proposed experiment, the agents never make any choices (also no delayed ones, as e.g., in Wheeler's "delayed choice" experiment[63]). Also,

none of the agents' statements refers to values that are no longer available at the time when the statement is made (cf. Table 3).

We conclude by suggesting a modified variant of the experiment, which may be technologically feasible. The idea is to substitute agents $\overline{F}$ and F by computers. Specifically, one would program them to carry out the tasks prescribed in Box 1, process the information accessible to them, and output statements such as "I am certain that W will observe $w = $ fail at 1:31." To account for the requirement that $\overline{F}$ and F's labs be isolated, one would need to ensure that the computers used for their simulation do not leak any information to their environment—a property which is necessarily satisfied by quantum computers. Such an experiment could then be used to verify the statements in Table 3. For example, aborting the experiment right after 1:13, one could, in the case when $z = +\frac{1}{2}$, read out statement $F^{1:13}$ made by agent F together with statement $\overline{F}^{1:02}$ that agent $\overline{F}$ has made just before. This would be a test for the correctness of statement $F^{1:13}$. Note that all statements in the fourth column of Table 3 could in the same way be tested experimentally. In this sense, quantum computers, motivated usually by applications in computing, may help us answering questions in fundamental research.

## Methods

**Information-theoretic description.** The experimental protocol described in Box 1 may be represented as a circuit diagram, Fig. 4. The diagram emphasises the information-theoretic aspects of the experiment. While all agents have full information about the overall evolution (the circuit diagram itself), they have access to different data (corresponding to different wires in the diagram).

**Derivation of statement $W^{0:00}$ using Assumption (Q).** In the analysis of the Gedankenexperiment we argued that the event $(\overline{w}, w) = (\overline{ok}, ok)$ must occur after finitely many rounds $n$, which is statement $W^{0:00}$ described shortly after (7). While this is a pretty obvious consequence of the Born rule, we now show that it already follows from Assumption (Q), which corresponds to the special case of the Born rule when it gives probability-1 predictions.

We consider Heisenberg operators relative to time $t_0 = 0{:}00$, i.e., right before the experiment starts. For any round $n$, let $W^n$ be the isometry from $\mathbb{C}$ to $\overline{L} \otimes L$ that includes the initialisation of system R in state $|init\rangle_R$ as well as $U_{R \to \overline{L}S}^{00 \to 10}$ and $U_{S \to L}^{10 \to 20}$ (cf. Table 1), i.e.,

$$W_{\mathbb{C} \to \overline{L}L}^n = (1_{\overline{L}} \otimes U_{S \to L}^{10 \to 20}) U_{R \to \overline{L}S}^{00 \to 10} |init\rangle. \tag{8}$$

The Heisenberg operator of the event $(\overline{w}, w) = (\overline{ok}, ok)$ in round $n$ relative to time $t_0$ can thus be written as

$$\pi_{(\overline{w},w)=(\overline{ok},ok)}^{(n)} = \left( W_{\mathbb{C} \to \overline{L}L}^n \right)^\dagger \left( |\overline{ok}\rangle\langle\overline{ok}|_{\overline{L}} \otimes |ok\rangle\langle ok|_L \right) \left( W_{\mathbb{C} \to \overline{L}L}^n \right). \tag{9}$$

We may now specify a Heisenberg operator $\pi_{halt}^{0:00}$ for the halting condition, i.e., that the event $(\overline{w}, w) = (\overline{ok}, ok)$ occurs in some round $n$,

$$\pi_{halt}^{0:00} = \sum_{n=0}^{\infty} \pi_{(\overline{w},w)=(\overline{ok},ok)}^{(n)} \prod_{m=0}^{n-1} \left( 1_{\mathbb{C}} - \pi_{(\overline{w},w)=(\overline{ok},ok)}^{(m)} \right). \tag{10}$$

Note that these are operators on $\mathbb{C}$, i.e., $\pi_{(\overline{w},w)=(\overline{ok},ok)}^{(n)} = p$ for some $p \in \mathbb{C}$. It follows directly from (7) that $p = 1/12 > 0$. We thus have

$$\pi_{halt}^{0:00} = \sum_{n=0}^{\infty} p(1-p)^n = 1_{\mathbb{C}}. \tag{11}$$

Inserting this measurement operator into the corresponding statement of Assumption (Q) yields statement $W^{0:00}$.

**Analysis within Bohmian mechanics.** According to Bohmian mechanics, the state of a system of particles consists of their quantum-mechanical wave function together with an additional set of variables that specify the particles' spatial positions.

While the wave function evolves according to the Schrödinger equation, the time evolution of the additional position variables is governed by another equation of motion, sometimes referred to as the "guiding equation". The general understanding is that these equations of motion must always be applied to the universe as a whole. As noted in ref. [32], "if we postulate that subsystems [rather than the universe] must obey Bohmian mechanics, we 'commit redundancy and risk inconsistency.'"

---

### Table 1 Time evolution

| Time interval within round $n$ | Time evolution of $\overline{F}$'s lab $\overline{L}$ | Time evolution of F's lab L |
|---|---|---|
| Before $n$:00 | set $R$ to $|init\rangle_R = \sqrt{1/3}|heads\rangle_R + \sqrt{2/3}|tails\rangle_R$ | [Irrelevant] |
| From $n$:00 to $n$:10 | $U_{R \to \overline{L}S}^{00 \to 10} = \begin{cases} |heads\rangle_R \mapsto |\overline{h}\rangle_{\overline{L}} \otimes |\downarrow\rangle_S \\ |tails\rangle_R \mapsto |\overline{t}\rangle_{\overline{L}} \otimes |\rightarrow\rangle_S \end{cases}$ | [Irrelevant] |
| From $n$:10 to $n$:20 | $U_{\overline{L} \to \overline{L}}^{10 \to 20} = 1_{\overline{L}}$ | $U_{S \to L}^{10 \to 20} = \begin{cases} |\downarrow\rangle_S \mapsto |-\frac{1}{2}\rangle_L \\ |\uparrow\rangle_S \mapsto |+\frac{1}{2}\rangle_L \end{cases}$ |
| From $n$:20 to $n$:30 | [Irrelevant] | $U_{L \to L}^{20 \to 30} = 1_L$ |

The two labs, $\overline{L}$ and L, are assumed to be isolated quantum systems. Technically, this means that their time evolution is described by norm-preserving linear maps, i.e., isometries. The second protocol step, for instance, in which F measures S, induces an isometry $U_{S \to L}^{10 \to 20}$ from S to L. The vectors $|-\frac{1}{2}\rangle_L$ and $|+\frac{1}{2}\rangle_L$ are defined as the outputs of this isometry, i.e., as the states of lab L at the end of the protocol step depending on whether the incoming spin was $|\downarrow\rangle_S$ or $|\uparrow\rangle_S$, respectively. For concreteness, one may think of them as states of the form (2)—although their structure is irrelevant for the argument. Analogously, $|\overline{h}\rangle_{\overline{L}}$ and $|\overline{t}\rangle_{\overline{L}}$ are defined as the states of lab $\overline{L}$ at the end of the first protocol step, depending on whether $r = $ heads or $r = $ tails, respectively

---

### Table 2 Measurements carried out by the agents

| Agent | Value | Measured system | Measurement completed at | Relevant vectors of measurement basis | Heisenberg projectors used for reasoning via (Q) |
|---|---|---|---|---|---|
| $\overline{F}$ | $r$ | R | $n$:01 | $|heads\rangle_R \quad |tails\rangle_R$ | $\pi_{w=ok}^{n:10} = \left[ (U_{S \to L}^{10 \to 20})^\dagger |ok\rangle_L \right] [\cdot]^\dagger$ <br> $\pi_{w=fail}^{n:10} = 1 - \pi_{w=ok}^{n:10}$ |
| F | $z$ | S | $n$:11 | $|\downarrow\rangle_S \quad |\uparrow\rangle_S$ | $\pi_{z=-\frac{1}{2}}^{n:10} = |\downarrow\rangle\langle\downarrow|_S$ <br> $\pi_{z=+\frac{1}{2}}^{n:10} = |\uparrow\rangle\langle\uparrow|_S$ |
| $\overline{W}$ | $\overline{w}$ | $\overline{L}$ | $n$:21 | $|\overline{ok}\rangle_{\overline{L}} = \sqrt{1/2}\left( |\overline{h}\rangle_{\overline{L}} - |\overline{t}\rangle_{\overline{L}} \right)$ | $\pi_{(\overline{w},z)=(\overline{ok},-\frac{1}{2})}^{n:00} = \left[ (U_{R \to \overline{L}S}^{00 \to 10})^\dagger |\overline{ok}\rangle_{\overline{L}} |\downarrow\rangle_S \right] [\cdot]^\dagger$ <br> $\pi_{(\overline{w},z)\neq(\overline{ok},-\frac{1}{2})}^{n:00} = 1 - \pi_{(\overline{w},z)=(\overline{ok},-\frac{1}{2})}^{n:00}$ |
| W | $w$ | L | $n$:31 | $|ok\rangle_L = \sqrt{1/2}\left( |-\frac{1}{2}\rangle_L - |+\frac{1}{2}\rangle_L \right)$ | $\pi_{(\overline{w},w)=(\overline{ok},ok)}^{n:00} = \left[ (U_{R \to \overline{L}S}^{00 \to 10})^\dagger (U_{S \to L}^{10 \to 20})^\dagger |\overline{ok}\rangle_{\overline{L}} |ok\rangle_L \right] [\cdot]^\dagger$ |

Each of the four agents observes a value, defined as the outcome of a measurement on a particular system at a particular time. The measurement basis vectors $|\overline{ok}\rangle_{\overline{L}}$ and $|ok\rangle_L$ shown in the last two rows are expressed in terms of states, such as $|-\frac{1}{2}\rangle$ and $|+\frac{1}{2}\rangle$, which are defined in Table 1. The last column shows the measurement operators that the agents insert into statement $A^{(ii)}$ when reasoning according to Assumption (Q). These operators are given in the Heisenberg picture, referring to the system's state at a particular time, which is specified by a superscript. The bracket $[\cdot]^\dagger$ stands for the adjoint of the preceding expression

**Table 3 The agents' observations and conclusions**

| Agent | Assumed observation | Statement inferred via (Q) | Further implied statement | Statement inferred via (C) |
|---|---|---|---|---|
| $\overline{F}$ | $r =$ tails at time $n{:}01$ | Statement $\overline{F}^{n:02}$ : "I am certain that W will observe $w =$ fail at time $n{:}31$." | | |
| F | $z = +\frac{1}{2}$ at time $n{:}11$. | Statement $F^{n:12}$ : "I am certain that $\overline{F}$ knows that $r =$ tails at time $n{:}01$." | Statement $F^{n:13}$ : "I am certain that $\overline{F}$ is certain that W will observe $w =$ fail at time $n{:}31$." | Statement $F^{n:14}$ : "I am certain that W will observe $w =$ fail at time $n{:}31$." |
| $\overline{W}$ | $w = \overline{ok}$ at time $n{:}21$ | Statement $\overline{W}^{n:22}$ : "I am certain that F knows that $z = +\frac{1}{2}$ at time $n{:}11$." | Statement $\overline{W}^{n:23}$ : "I am certain that F is certain that W will observe $w =$ fail at time $n{:}31$." | Statement $\overline{W}^{n:24}$ : "I am certain that W will observe $w =$ fail at time $n{:}31$." |
| W | announcement by agent $\overline{W}$ that $w = \overline{ok}$ at time $n{:}21$ | Statement $W^{n:26}$ : "I am certain that $\overline{W}$ knows that $w = \overline{ok}$ at time $n{:}21$" | Statement $W^{n:27}$ : "I am certain that $\overline{W}$ is certain that I will observe $w =$ fail at time $n{:}31$." | Statement $W^{n:28}$ : "I am certain that I will observe $w =$ fail at time $n{:}31$." |

The statements that the individual agents can derive from quantum theory depend on the information accessible to them (cf. Fig. 2). Agent $\overline{F}$, for instance, if she observes $r =$ tails, can use this information to infer $w$, which will later be observed and announced by W

**Table 4 Interpretations of quantum theory**

| | (Q) | (S) | (C) |
|---|---|---|---|
| Copenhagen | ✓ | ✓ | × |
| HV theory applied to subsystems | ✓ | ✓ | × |
| HV theory applied to entire universe | × | ✓ | ✓ |
| Many worlds | ? | × | ? |
| Collapse theories | × | ✓ | ✓ |
| Consistent histories | ✓ | ✓ | × |
| QBism | ✓ | ✓ | × |
| Relational quantum mechanics | ✓ | ✓ | × |
| CSM approach | × | ✓ | ✓ |
| ETH approach | × | ✓ | ✓ |

The proposed Gedankenexperiment can be employed to study various interpretations of quantum theory. Theorem 1 implies that each of them must violate at least one of the Assumptions (Q), (C), and (S) (indicated by ×). For hidden variable (HV) theories, it is relevant whether agents who are using the theory apply its laws (e.g., the guiding equation in the case of Bohmian mechanics) to subsystems around them or to the universe as a whole.

The Gedankenexperiment presented in this work shows that this risk is real. Indeed, if the agents applied the Bohmian equations of motion directly to the relevant systems around them, rather than to the universe as a whole, their reasoning would be the same as the one prescribed by (Q). But since Bohmian mechanics also satisfies (S), this would, by virtue of Theorem 1, imply a violation of (C), i.e., the agents' conclusions would contradict each other. (This finding should not be confused with the known fact that, if the spatial position of a particle is measured, the Bohmian position of the measurement device's pointer is sometimes incompatible with the Bohmian position of the measured particle[64–68].)

The directive in ref. [32] that Bohmian mechanics should be applied to the entire universe means that the agents must model themselves from an outside perspective. This ensures that they all have the same view, so that reasoning according to (C) is unproblematic. But then, because of Theorem 1, (Q) is necessarily violated. This is indeed confirmed by an explicit calculation in Bohmian mechanics, which reveals that statement $\overline{F}^{n:02}$ of Table 3 does not hold there. Furthermore, the time order of the measurements carried out by agents $\overline{W}$ and W is relevant within Bohmian mechanics. If agent W measured before agent $\overline{W}$ then, according to Bohmian mechanics, statement $\overline{W}^{n:22}$ would be invalid whereas $\overline{F}^{n:02}$ would hold. This is a clear departure from standard quantum mechanics, where the time order in which agents $\overline{W}$ and W carry out their measurements is irrelevant, because they act on separate systems.

This violation of (Q) raises the question under what circumstances Bohmian mechanics still endorses the use of the quantum-mechanical Born rule for predicting the outcome of a measurement. A candidate criterion could be that such a prediction is only valid if a memory of the prediction is available upon completion of the measurement. One may then be tempted to argue that agent $\overline{F}$'s statement $\overline{F}^{n:02}$, for instance, is invalid because $\overline{F}$ is herself subject to a measurement, which may destroy her memory of the prediction for $w$ before that value is measured. This argument does however not work. The reason is that, in the relevant case when $w = \overline{ok}$, the value $r$ and hence also agent $\overline{F}$'s prediction for $w$ is, by virtue of statements $\overline{W}^{n:22}$ and $F^{n:12}$, retrievable at the time when $w$ is measured.

**Analysis within the CH formalism.** In the CH formalism, statements about measurement outcomes are phrased in terms of "histories". These must, by definition, be elements of a whole family of histories, called a "framework", that satisfies certain consistency conditions. In the Gedankenexperiment proposed in this work, a possible history would be

*History $h_1$:* "In round $n$ the outcomes $r =$ tails, $z = +\frac{1}{2}$, $\overline{w} = \overline{ok}$, and $w =$ ok were observed."

To verify that $h_1$ is indeed a valid history, one has to construct a framework containing this history. It is straightforward to check that one such framework is the set consisting of $h_1$ together with the additional histories

*History $h_2$:* "In round $n$ the outcomes $r =$ tails, $z = +\frac{1}{2}$, $\overline{w} = \overline{ok}$, and $w =$ fail were observed."

*History $h_3$:* "In round $n$ the outcomes $r =$ heads, $z = +\frac{1}{2}$, and $\overline{w} = \overline{ok}$ were observed."

*History $h_4$:* "In round $n$ the outcomes $z = -\frac{1}{2}$, and $\overline{w} = \overline{ok}$ were observed."

*History $h_5$:* "Outcome $\overline{w} = \overline{fail}$ was observed."

The CH formalism contains the Born rule as a special case and hence fulfils Assumption (Q). Since it also satisfies (S), it follows from Theorem 1 that it violates (C). To illustrate how this violation manifests itself, we may consider a shortened version of history $h_1$, which leaves the values $z$ and $\overline{w}$ unmentioned:

*History $h'_1$:* "In round $n$ the outcomes $r =$ tails and $w =$ ok were observed."

The CH formalism provides a rule to assign probabilities to these histories, which turn out to be

$$Pr[h_1] = \frac{1}{12} \quad \text{and} \quad Pr[h'_1] = 0 . \tag{12}$$

Note that these probabilities disagree with the fact that $h'_1$ is just a part of history $h_1$, i.e., $h_1 \Rightarrow h'_1$. (This finding may be compared to the "three box paradox"[69], where calculations in three different consistent frameworks yield mutually incompatible probability assignments; see Section 22 of ref. [35] as well as [70] for a discussion.)

The CH formalism accounts for this disagreement by imposing the rule that logical reasoning must be constrained to histories that belong to a single framework, which is not the case for $h_1$ and $h'_1$. To illustrate what this means, it is useful to return to the casino example described in the Discussion section above. Within a framework that contains history $h'_1$, the gambler's reasoning is correct, for $Pr[h'_1] = 0$. That is, $w =$ ok implies that $r =$ heads. Conversely, considering the framework above, which contains history $h_1$, it is readily verified that the other histories, $h_2$–$h_5$, have probabilities $\frac{1}{12}$, 0, 0, and $\frac{5}{6}$, respectively. That is, all nonzero probability histories of this framework that agree with the observation $\overline{w} = \overline{ok}$ also assert that $z = +\frac{1}{2}$ and $r =$ tails. This seems to be in agreement with the casino's argument, i.e., $\overline{w} = \overline{ok}$ implies that $z = +\frac{1}{2}$ and $r =$ tails. However, because the framework does not include a history that talks about $r$ alone, it disallows the—seemingly obvious—implication $\left(r = \text{tails and } z = +\frac{1}{2}\right) \Rightarrow r =$ tails. In other words, within the CH formalism, the casino can prove that $r =$ tails and $z = +\frac{1}{2}$, but not that $r =$ tails.

**Analysis within QBism.** QBism is one of the most far-reaching subjectivistic interpretations of quantum mechanics. It regards quantum states as representations of an agent's personal knowledge, or rather beliefs, about the outcomes of future measurements, and it also views these outcomes as personal to the agent.

To reflect these tenets of QBism in the analysis of the Gedankenexperiment, it is useful to imagine that the agents write their observations and conclusions into a personal notebook. For example, according to Table 3, when agent $\overline{F}$ gets $r =$ tails in round $n = 1$, she may put down the following

*Statement $\overline{F}^{1:02}$:* "$r =$ tails at time 1:01, hence I am certain that we will hear W announcing $w =$ fail at the end of this round."

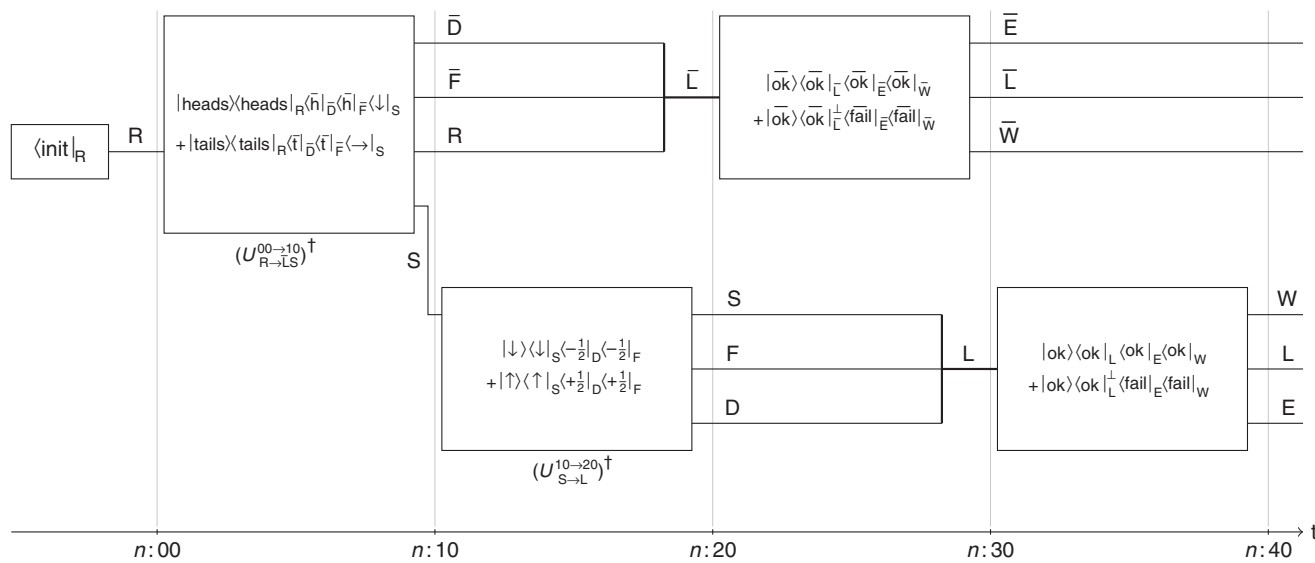

**Fig. 4** Circuit diagram representation of the Gedankenexperiment. The actions of the agents during the protocol correspond to isometries (boxes) that act on particular subsystems (wires). For example, the measurement of S by agent F in the second protocol step, which starts at time $n{:}10$, induces an isometry $U_{S \to L}^{10 \to 20}$ from S to F's lab L, analogous to the one defined by (2). The subsystems labelled by $\bar{F}$, F, $\bar{W}$, and W contain the agents themselves. Similarly, $\bar{D}$, D, $\bar{E}$, and E are "environment" subsystems, which include the agents' measurement devices. The states of these subsystems depend on the measurement outcome, which is indicated by their label. For example, $\left| +\frac{1}{2} \right\rangle_F$ is the state of F when the agent has observed $z = +\frac{1}{2}$

Here the phrase "is certain that" expresses a degree of belief and may also be replaced by something like "would bet an arbitrarily large amount on". Similarly, agent F, when she gets $z = +\frac{1}{2}$, may write into her notebook

Statement F[1:12]:"$z = +\frac{1}{2}$ at time 1:11, hence I am certain that, if I now checked $\bar{F}$'s notebook, I would read that $r =$ tails at time 1:01."

Agent F may as well write about agent $\bar{F}$'s conclusions, i.e.,

Statement F[1:13]:"$z = +\frac{1}{2}$ at time 1:11, hence I am certain that, if I now checked $\bar{F}$'s notebook, I would read that she is certain that we will hear W announcing $w =$ fail at the end of this round."

Agent F may now be tempted to conclude from the above that

Statement F[1:14]:"$z = +\frac{1}{2}$ at time 1:11, hence I am certain that we will hear W announcing $w =$ fail at the end of this round."

However, permitting such implications is akin to assuming (C). Because QBism satisfies (Q) and (S), it would result in the agents issuing contradictory statements. The Gedankenexperiment is thus an example of a multi-agent scenario where, to ensure consistency of QBism, implications of the type F[1:13] ⇒ F[1:14] must be disallowed. Nevertheless, there should be ways for agents to consistently reason about each other. One may therefore ask whether (C) could be substituted by another (weaker) rule that enables such reasoning but does not lead to contradictions. This question is currently being investigated (J.B. DeBrota, C.A. Fuchs, and R. Schack, manuscript in preparation).

## Data availability
No data sets were generated or analysed during the current study.

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

## Acknowledgements

We would like to thank Yakir Aharonov, Mateus Araújo, Alexia Auffèves, Jonathan Barrett, Veronika Baumann, Serguei Beloussov, Charles Bennett, Hans Briegel, Časlav Brukner, Harry Buhrman, Adán Cabello, Giulio Chiribella, Roger Colbeck, Patricia Contreras Tejada, Giacomo Mauro D'Ariano, John DeBrota, Lídia del Rio, David Deutsch, Artur Ekert, Michael Esfeld, Philippe Faist, Aaron Fenyes, Hugo Fierz, Jürg Fröhlich, Christopher Fuchs, Shan Gao, Svenja Gerhard, Edward Gillis, Nicolas Gisin, Sheldon Goldstein, Sabrina Gonzalez Pasterski, Gian Michele Graf, Philippe Grangier, Bob Griffiths, Arne Hansen, Lucien Hardy, Aram Harrow, Klaus Hepp, Paweł Horodecki, Angela Karanjai, Adrian Kent, Gijs Leegwater, Matthew Leifer, Seth Lloyd, John Loverain, Markus Müller, Thomas Müller, Hrvoje Nikolić, Travis Norsen, Nuriya Nurgalieva, Jonathan Oppenheim, Sandu Popescu, Matthew Pusey, Gilles Pütz, Joseph Renes, Jess Riedel, Valerio Scarani, Rüdiger Schack, Robert Spekkens, Cristi Stoica, Antoine Suarez, Tony Sudbery, Stefan Teufel, Roderich Tumulka, Lev Vaidman, Vlatko Vedral, Mordecai Waegell, Andreas Winter, Stefan Wolf, Filip Wudarski, Christa Zoufal, and Wojciech Zurek for comments and discussions. This project was supported by the Swiss National Science Foundation (SNSF) via the National Centre of Competence in Research "QSIT", by the Kavli Institute for Theoretical Physics (KITP) at the University of California in Santa Barbara, by the Stellenbosch Institute for Advanced Study (STIAS) in South Africa, by the US National Science Foundation (NSF) under grant No. PHY17-48958, by the European Research Council (ERC) under grant No. 258932, and by the European Commission under the project "RAQUEL".

## Author contributions

D.F. and R.R. made equally substantial contributions to this work.

## Additional information

**Competing interests:** The authors declare no competing interests.

