## [Peer Review File · Nature Communications]

Reviewers' comments:

Reviewer #1 (Remarks to the Author):

In this paper, the authors combine the idea of Wigner's friend with Hardy's-paradox-type reasoning to prove that "quantum theory cannot consistently describe itself". In more detail, the authors construct an elaborate thought experiment that proves unambiguously, in the form of a no-go theorem, that three seemingly innocent physical assumptions cannot all hold true in all circumstances. The three assumptions are, in a nutshell:

(Q): If the Born rule assigns probability 1 to a proposition, then any agent can be certain that this proposition holds.

(C): We have a form of "consistency between observers" that allows an observer to obtain conclusions based on reasoning about the conclusions of other observers.

(S): At any time, there is a meaningful way to say that EITHER a proposition OR its negation is true, but not both at the same time in some sense (the latter would apply, for example, to many-worlds-like interpretations).

The results and conclusions are certainly novel and of interest to others in the community and the wider field. More than that, I regard this result as major progress in the foundations of quantum mechanics and of broad interest to all physicists. I would put its significance above the recent Pusey-Barrett-Rudolph result (which has sparked immense interest in the community and beyond), and see great potential that this paper will influence thinking in the foundations of quantum mechanics for years to come.

This paper has been discussed very lively in the quantum information and foundations community since its first version has appeared on the arXiv. It is clearly visible that the authors have invested a lot of effort into incorporating all insights from these discussions into the paper. This version of the paper is extremely clearly written and contains both technically and conceptually very deep and sophisticated arguments and insights. For example, the presentation in Section IV (on the different interpretations of quantum mechanics) is very detailed and competent, and has obviously grown into this mature form over the course of many months and over discussions with many people that represent the various interpretations.

As far as I can see (and I have been going through the argument many times over the last year), all calculations and arguments are correct. Moreover, the paper is very well written. I strongly endorse publication of this paper in Nature Communications.

I do have a few minor comments. These comments should be easy to address, and they do not alter the conclusion that this is an excellent paper as it stands.

* Figure 4 should be improved to be better visible (lighter blue, bigger fonts etc.).

* When the "Experimental Protocol" is introduced, it should be explained in a short sentence that "@n:00: denotes the times at the n-th run (it's a bit confusing first).

* In IV.C ("Theories that violate C"), the last sentence says: "... implies that predictions obtained from HV models are generally also inconsistent."

This terminology should be avoided (probably this is a left-over from the older version) because it is misleading. It is not that such a theory would be "inconsistent" as a (mathematical/physical) theory, but that assumption (C) would be violated. And (C) is not simply "consistency", but a certain type of "multiple-observer consistency" (or however the authors would like to name it).

* Last few sentences of Section IV:

"...Theorem 1 is independent of how probabilities are interpreted." It might make sense to admit (say, in a short footnote) that assumption (Q) sneaks in a partial interpretation of probability, by saying that overlap 1 in the QM formalism implies that the corresponding event definitely

happens.

"It avoids the [...] assumption that [...] outcomes obtained by different agents simultaneously have well-defined values."

Optional request (I leave it to the authors to answer this or not): but by applying Assumption (C), we logically relate the outcomes of the different agents, and in this sense treat them as parts of a single context. Thus, isn't Assumption (C) more or less equivalent to (or a certain form of) "simultaneous well-definedness"?

* Last paragraph of V. Discussion ("We conclude by suggesting a modified variant of the experiment,..."): I think a clarification is in place. Namely, if this is supposed to give the idea for an experiment (with agents replaced by quantum computer, which in itself is a good point of course), then what exactly should the experimenter do/try to verify? What would be the hypothesis that is confirmed/rejected by the experiment?

Isn't it simply that case that whatever the experiment is, we would have to use quantum theory from the point of view of the external physicist that builds the experiment and publishes the result in PRL, in which case there would be no ambiguity? Or is the idea here to operationalize the thought experiment in a way that decides between the various interpretations of quantum theory (giving different predictions in this case)?

Reviewer #2 (Remarks to the Author):

In this work, the authors consider a particular Gedanken experiment involving a nested situation wherein several agents apply a theory to reason about systems which contain other agents who themselves also use the same theory to make prediction on smaller systems. They show that, assuming that quantum theory can be used to describe such situation, i.e., "Assumption (Q)" of the manuscript, combined with a plausible assumption of nested reasoning, "Assumption (C)", and that each agent can only get a single definite outcome in each run of measurement, "Assumption (S)", may lead to contradictory statements on the measurement outcomes drawn by different agents. They thereby arrive at a no-go theorem stating that any theory which claims to describe the above Gedanken experiment must at least violate one of the above three assumptions: (Q), (C), and (S).

To arrive at the conclusion, they extended the well-known Wigner Gedanken experiment, wherein an agent W uses quantum theory to describe a closed system L which contains another agent F who uses quantum theory to describe a smaller system. While quantum theory can be applied to describe Wigner's Gedanken experiment consistently, they show that a certain type of complication does lead to a contradiction. To do so, they introduce another agent W' (any overline in the original manuscript is replaced here with a prime) who uses quantum theory to describe a different closed system L' which contains yet another agent F' . In each round of the experiment, agent F' first flips a specific biased quantum coin (quantum random generator) with outcome $r = (\text{heads}, \text{tails})$, and sends an electron to agent F with a spin state that depends on the outcome of the quantum coin. Agent F then makes a spin measurement on his electron w.r.t. the $(|\uparrow\rangle, |\downarrow\rangle)$ basis with outcome $z = (-1/2, 1/2)$; agent W makes a measurement on system L w.r.t. a specific basis with measurement outcome denoted $w = (\text{ok}, \text{fail})$, and agent W' on L' w.r.t. a specific basis with measurement outcome $w' = (\text{ok}', \text{fail}')$. The experiment is repeated until agent W announces that " $w = \text{ok}$ " and agent W' announces that " $w' = \text{ok}'$ ". The authors show that if all the agents use quantum mechanics to reason about the outcomes of the measurements of the other agents based on their measurement outcomes, assuming that they may use nested reasoning, i.e., Assumption (C), they may end up with contradictory statements.

The presentation of this argument is very hard to follow. There are many undefined notation such as " $n:20$ " and " $n:30$ " and unmotivated symbols. Certainly the presentation could and should be much improved, so that the reader has hope of eventually understanding the authors' ideas.

For later discussion, let us try to summarize what the authors have done in their analysis. First, using quantum mechanics, i.e., Assumption (Q), they derived the following statements:

(P1): If "w' = ok' " ---then- "z = 1/2"

(P2): If "z = 1/2" ---then- "r = tail"

(P3): If "r = tail" ---then- "w = fail"

(P4): There is a round of experiment which yields "w=ok" and "w' = ok' ".

Statements (P1), (P2), (P3), and (P4) above correspond respectively to $s_Q^{\overline{W}}$, s_Q^F , $s_Q^{\overline{F}}$, and s_Q^W of the manuscript.

Granted the validity of nested reasoning, Assumption (C), the statements (P1), (P2) and (P3) directly imply:

(P5) If "w' = ok' " ---then- "w = fail".

Clearly, statement (P5) contradicts statement (P4).

The authors also show, in the Discussion, that the above statements lead to a dispute (contradiction) between the conclusions drawn based on two methods of retrodiction applied by a gambler (agent W) and employees of a casino (agents W', F and F'). Assume that statement (P4) is valid so that there is a round in which "w = ok" and "w' = ok' ". Then, from "w = ok" and statement (P3), we have "r = head". However, from "w' = ok'" and statements (P1) and (P2), we have "r = tail".

The authors then discuss how different interpretations and modifications of quantum mechanics deal with the above results by identifying which assumptions, (Q), (C), (S) are correspondingly violated. This discussion leads to a natural categorization. They conclude, as emphasized in the title of the manuscript, that quantum mechanics cannot be used to describe itself.

The claim of the authors, if correct, certainly may have wide implications, in our attempt to better understand the meaning and scope of quantum theory; it may suggest fresh insight to reconstruct quantum mechanics from physical axioms, and may have practical applications in quantum information and perhaps also in quantum cosmology. Moreover, the scheme wherein they derive the contradiction using the extension of Wigner's Gedanken experiment is novel, and may suggest fresh ideas in quantum information.

However, in our opinion, to arrive at their conclusion implied by the title, certain conceptual issues have to be clarified. First, when extending from the Wigner Gedanken experiment to their set-up, they do not only put in two more agents, but they also introduced more (complicated) measurements. In particular, their Gedanken experiment now involves a pair of incompatible measurements (see below). On the other hand, it is known that in quantum mechanics in general it is impossible to combine the outcomes of incompatible measurements into a single coherent story (complementarity). See for example Y.-C. Liang, R. W. Spekkens, and H. M. Wiseman, Phys. Rep. 506, 1–39 (2011), wherein it is elaborated that quantum mechanics in general does not allow transitivity of implications, connecting statements which are obtained from incompatible measurements. Applying this to their Gedanken experiment, we have to be careful when, given statements (P1), (P2), and (P3), we conclude with (P5), which is otherwise allowed if Assumption (C) is valid. In other words, within quantum mechanics, Assumption (C) is not allowed in more general situations than that considered by the authors.

To be more concrete, consider the measurement performed by agent W' and agent W. In their analysis of measurement, the authors use the Heisenberg picture relative to time $t=10$, so that the relevant quantum state for both measurements is given by Eq. (6). It is then clear that the two measurements leading, respectively, to statement (P1) and (P4), are incompatible: namely, the associated projectors (given respectively in line 113 and 118 on page 6) do not commute. Statement (P1) is obtained by measurement of the spin of the electron w.r.t. the basis

($|\uparrow\rangle, |\downarrow\rangle$), while statement (P4) is obtained by measurement of the spin of the electron w.r.t. the basis ($|\rightarrow\rangle, |\leftarrow\rangle$). The two measurements cannot therefore be done simultaneously relative to time $t=10$ in the Heisenberg picture. This means that the records of measurement of agent W' at $t=30$ and W at $t=40$ are not compatible; they refer to two incompatible experimental set-ups (invoking two measurement devices which are incompatible with each other). The same analysis applies to the gambler-casino dispute. The casino's employees, using statements (P1), cannot make inference, based on the truth assignment given by statement (P4) since they are obtained with incompatible measurements.

It seems also clear from their analysis that the statements (P1) and (P4) above are obtained as if agent W' make two measurements of L' w.r.t. basis ($|ok'\rangle, |fail'\rangle$) with two different and incompatible contexts: in one context the basis ($|\uparrow\rangle, |\downarrow\rangle$) is used for the spin measurement of electron simultaneously with the measurement of L' , and in the other context the basis ($|\rightarrow\rangle, |\leftarrow\rangle$) is being used. In other words, we think that in their analysis, the authors have committed counterfactual reasoning and noncontextual truth-value assignment: agent W , at time $t=40$, makes his/her inference (using quantum mechanics) with the assumption that the measurement made by agent W' earlier at $t=30$ did not change the system, even if their measurements are incompatible with each other. This has to be further clarified. For this reason, in view of the fact that they use a similar type of contradiction, we also think that more detailed discussion is needed to clarify its relation with the Hardy' paradox that they cited (Refs. [10,11]).

Hence, in our opinion, the authors have to make their arguments clearer to identify the crucial factor which leads to the contradiction in their analysis of the extended Wigner Gedanken experiment: what new ingredients introduced to the Wigner Gedanken experiment lead to the contradiction? Whether the contradiction has anything to do with the assumption that quantum mechanics is used to describe a system which contains an agent who also uses quantum mechanics, a self-referential use of quantum theory, as emphasized in the title of their manuscript? Or, whether the contradiction arises due to a general impossibility of transitivity of implications, involving statements which are obtained using incompatible measurements, in other words whether it is the use of counterfactual reasoning and noncontextual assignment of truth-value which leads to the contradiction? Even if it turns out to be the latter, their set-up to derive the contradiction is novel, and may have applications in quantum information as suggested by their discussion in casino-gambler. However, for the manuscript to be further considered, we suggest that the authors clarify the above issues.

Minor suggestions and questions:

- We think the manuscript will be more easily readable, and the contradictions may easily be seen, if the authors express the statements implied by Assumption (Q) pictorially/graphically in term of arrows of implications.

- In their article on page 9 line 177, they argue that, as the implication of the no-go theorem that they derived, Bohmian mechanics (as a theory of the universe), must satisfy Assumption (C), thus allowing nested reasoning. However, Bohmian mechanics is a contextual ontological model so that, based on our reading of the Gedanken experiment discussed above, it should violate Assumption (C). They also claim on page 13 line 307 that, instead, Bohmian mechanics violates Assumption (Q), namely it makes predictions that contradict quantum mechanics. To avoid confusion, we think that the discussion of Bohmian mechanics with regard to the above issue needs further clarification.

- It does not seem to be clearly explained within the manuscript the reason why W' and W carry out their measurements at two different times. Does the time order matter?

- Is the assumption (S) necessary? Even in a many-worlds interpretation, it seems clear to us that each observer in each world only sees one outcome.

Reviewer #3 (Remarks to the Author):

This manuscript proves if quantum mechanics is universally applicable and measurement outcomes are objective then there are situations where the predictions for those outcomes are inconsistent. I would argue that this theorem stands in relation to the original Wigner's friend thought experiment roughly as Bell's theorem stands in relation to the EPR paradox. The EPR paradox showed that quantum mechanics on its own cannot give a locally causal description of reality, whereas Bell's theorem showed that such a description cannot even be achieved by adding additional variables. Likewise this theorem shows that additional variables cannot always resolve the sort of disagreements Wigner has with the friend.

I think this is a very important result in quantum foundations, with implications for almost every approach to interpreting quantum mechanics. There is a good discussion of some of these implications in the paper, and it will be interesting to see the responses from proponents of the various interpretations. On the other hand, although theorems about the foundations of quantum mechanics have often found application in quantum information, that seems unlikely in this case since there is no reason to think that performing the experiment on actual people will ever be feasible. On balance I think this manuscript will be of wide interest and therefore support its publication in Nature Communications.

The paper is generally well written, if a little verbose in places. A few minor comments:

p2, l56: naturally -> natural

p3, l69: The phase in the superposition of $|\text{heads}\rangle$ and $|\text{tails}\rangle$ are specified "for completeness", which suggests that it doesn't matter what it is. But I think the statement on p6, l113 wouldn't be true with a different phase.

p3, eq6: Hasn't F's measurement happened by time $n:10$, so that there would be entanglement with F? This looks more like the state at, say, $n:05$.

p10: I'm sceptical of the claim that the experiment would be feasible if quantum computers played the role of the friends. Either the term "quantum computer" includes the final measurements used to obtain the result of the computation, in which case the W measurements would be practically impossible as usual, or the term only refers to the unitary circuit, in which case I don't see any reason to think the computer obtains outcomes.

p13: It would be nice if there were some details of the calculations in the Bohmian and Consistent Histories cases, space permitting.

General Changes

- We have restructured the document to comply with the guidelines. (The intro still contains references to figures, but because they are clearly introductory we hope that this is fine.)
- We have removed all footnotes as requested. Whenever suitable, we have included the corresponding information in the main text.
- We have slightly rephrased Assumption (S) to emphasise the point that we do not demand that an agent can generally assign values to outcomes of measurements by other agents. Only *if* an agent can assign a value to an outcome then (S) disallows that he/she also also assigns an opposite value to this outcome.
- We have replaced the former Table I by a pair of tables, which contain more complete specifications of the elements that are needed to define the thought experiment.
- We have added a short section to acknowledge funding and put the data availability statement as requested.

Response to Referees

We thank the reviewers for their careful reading of the manuscript and for their constructive comments.

Reviewer #1

Figure 4 should be improved to be better visible (lighter blue, bigger fonts etc.).

We have redrawn the figure and hope that its elements are now better readable.

When the “Experimental Protocol” is introduced, it should be explained in a short sentence that “@n:00” denotes the times at the n-th run (it’s a bit confusing first).

Done. (This is now said in the sentence just underneath the box.)

In IV.C (“Theories that violate C”), the last sentence says: “... implies that predictions obtained from HV models are generally also inconsistent.” This terminology should be avoided (probably this is a left-over from the older version) because it is misleading. It is not that such a theory would be “inconsistent” as a (mathematical/physical) theory, but that assumption (C) would be violated. And (C) is not simply “consistency”, but a certain type of “multiple-observer consistency” (or however the authors would like to name it).

Indeed, the term “inconsistency” was not meant to refer to the theory itself. Rather, there does not exist an assignment of values to the HVs that is consistent with the conclusions of all agents.

Last few sentences of Section IV: “... Theorem 1 is independent of how probabilities are interpreted.” It might make sense to admit (say, in a short footnote) that assumption (Q) sneaks in a partial interpretation of probability, by saying that overlap 1 in the QM formalism implies that the corresponding event definitely happens.

We have added such a statement in the part on “implicit assumptions” at the end of the section (which is now Section II.C).

“It avoids the [...] assumption that [...] outcomes obtained by different agents simultaneously have well-defined values.” Optional request (I leave it to the authors to answer this or not): but by applying Assumption (C), we logically relate the outcomes of the different agents, and in this sense treat them as parts of a single context. Thus, isn’t Assumption (C) more or less equivalent to (or a certain form of) “simultaneous well-definedness”?

As far as we can see, Assumption (C) is strictly weaker than simultaneous well-definedness. For example, in the original Wigner’s friend experiment, (C) does not force W to assign a definite value to the outcome z observed by F. We have added a bracket to mention this in the part on “implicit assumptions.”

Last paragraph of V. Discussion (“We conclude by suggesting a modified variant of the experiment, ...”): I think a clarification is in place. Namely, if this is supposed to give the idea for an experiment (with agents replaced by quantum computer, which in itself is a good point of course), then what exactly should the experimenter do/try to verify? What would be the hypothesis that is confirmed/rejected by the experiment? Isn’t it simply that case that whatever the experiment is, we would have to use quantum theory from the point of view of the external physicist that builds the experiment and publishes the result in PRL, in which case there would be no ambiguity? Or is the idea here to operationalize the thought experiment in a way that decides between the various interpretations of quantum theory (giving different predictions in this case)?

This is a very good point (or question). Our idea was that the experiment could test the implications of Assumption (Q) that are relevant to our argument, i.e., the ones listed in Table II. Although such a test would require additional assumptions, it would address concerns that have been raised about the physical plausibility of these statements (see, e.g., arXiv:1802.06396, where it is argued that one of the statements of Table II must be wrong).

We have slightly expanded this part of the discussion to clarify our intended purpose of such experiments.

Reviewer #2

There are many undefined notation such as “n:20” and “n:30” and unmotivated symbols.

We have added explanations (see, e.g., the description of the experimental protocol) and omitted the symbol @, which was unnecessary. We have also largely rewritten the part with the more formal description and analysis of the experimental protocol, simplified the notation used there, and added a table with the relevant definitions (Table I).

However, in our opinion, to arrive at their conclusion implied by the title, certain conceptual issues have to be clarified. [...] To be more concrete, consider the measurement performed by agent W’ and agent W. In their analysis of measurement, the authors use the Heisenberg picture relative to time $t=10$, so that the relevant quantum state for both measurements is given by Eq. (6). It is then clear that the two measurements leading, respectively, to statement (P1) and (P4), are incompatible: namely, the associated projectors (given respectively in line 113 and 118 on page 6) do not commute. Statement (P1) is obtained by measurement of the spin of the electron w.r.t. the basis

($|\uparrow\rangle, |\downarrow\rangle$), while statement (P4) is obtained by measurement of the spin of the electron w.r.t. the basis ($|\rightarrow\rangle, |\leftarrow\rangle$). The two measurements cannot therefore be done simultaneously relative to time $t=10$ in the Heisenberg picture. This means that the records of measurement of agent W' at $t=30$ and W at $t=40$ are not compatible; they refer to two incompatible experimental set-ups (invoking two measurement devices which are incompatible with each other).

We do not think that the two measurements, the one by agent W' and the one by agent W , are incompatible. The reason is that the measurement of W' (\bar{W} in our notation) acts on lab L' (\bar{L} in our notation), whereas the measurement of W acts on lab L (see the description of the experimental protocol on page 3). These two systems, L' and L , are separate from each other (they are even at different spacial locations). We hope that these aspects are clarified by the new Figure 4.

It seems also clear from their analysis that the statements (P1) and (P4) above are obtained as if agent W' make two measurements of L' w.r.t. basis ($|ok'\rangle, |fail'\rangle$) with two different and incompatible contexts: in one context the basis ($|\uparrow\rangle, |\downarrow\rangle$) is used for the spin measurement of electron simultaneously with the measurement of L' , and in the other context the basis ($|\rightarrow\rangle, |\leftarrow\rangle$) is being used. In other words, we think that in their analysis, the authors have committed counterfactual reasoning and noncontextual truth-value assignment: agent W , at time $t=40$, makes his/her inference (using quantum mechanics) with the assumption that the measurement made by agent W' earlier at $t=30$ did not change the system, even if their measurements are incompatible with each other. This has to be further clarified. For this reason, in view of the fact that they use a similar type of contradiction, we also think that more detailed discussion is needed to clarify its relation with the Hardy' paradox that they cited (Refs. [10,11]).

We agree with the reviewer that the measurement made by agent W' at time $t=n:30$ could change system L' . However, in our analysis, we never make the assumption that the system L' (which is the one measured by W') remains unchanged. Unfortunately, we were not able to figure out why the referee thinks that we are making this assumption.

Hence, in our opinion, the authors have to make their arguments clearer to identify the crucial factor which leads to the contradiction in their analysis of the extended Wigner Gedanken experiment: what new ingredients introduced to the Wigner Gedanken experiment lead to the contradiction? Whether the contradiction has anything to do with the assumption that quantum mechanics is used to describe a system which contains an agent who also uses quantum mechanics, a self-referential use of quantum theory, as emphasized in the title of their manuscript? Or, whether the contradiction arises due to a general impossibility of transitivity of implications, involving statements which are obtained using incompatible measurements, in other words whether it is the use of counterfactual reasoning and noncontextual assignment of truth-value which leads to the contradiction? Even if it turns out to be the latter, their set-up to derive the contradiction is novel, and may have applications in quantum information as suggested by their discussion in casino-gambler. However, for the manuscript to be further considered, we suggest that the authors clarify the above issues.

The question of what exactly causes the contradiction is of course an important one. However, we do not think that it is possible to answer it unless one admits a particular interpretation (such as "consistent histories"). Although one can identify the cause of the contradiction within some of the interpretations (see Section II.C and the Methods section in the revised document), the different interpretations yield differing conclusions, i.e., there does not seem to exist a general answer.

Because of this, our main result just says that certain assumptions, namely (Q), (S), and (C),

when taken together, lead to a contradiction, but it does not further localise the problem. We note that the same is true for other no-go results, such as Bell’s theorem. The latter also asserts that a certain set of assumptions (e.g., the correctness of quantum theory, local causality, and freedom of choice) are contradictory, but does not declare one single of them to be the cause of the contradiction.

Having said this, we would like to stress that, in our thought experiment, the contradiction does not arise from counterfactual reasoning. In fact, the agents never make any choices, i.e., their measurements are always the same. They can therefore, in particular, not reason about what would have happened if they measured differently.

To clarify these points, we have expanded the discussion section in the revised document. We have in particular included a paragraph that discusses the relation to other no-go results.

Minor suggestions and questions: We think the manuscript will be more easily readable, and the contradictions may easily be seen, if the authors express the statements implied by Assumption (Q) pictorially/graphically in term of arrows of implications.

We decided to express the conclusions in Table II in the form of text. The reason is that an arrow suggests an implication that is generally valid. However, an important aspect of our analysis is that all statements can be understood as subjective, i.e., they express agent-specific conclusions. This is why we prefer to use the text form “... is certain that ...”

In their article on page 9 line 177, they argue that, as the implication of the no-go theorem that they derived, Bohmian mechanics (as a theory of the universe), must satisfy Assumption (C), thus allowing nested reasoning. However, Bohmian mechanics is a contextual ontological model so that, based on our reading of the Gedanken experiment discussed above, it should violate Assumption (C). They also claim on page 13 line 307 that, instead, Bohmian mechanics violates Assumption (Q), namely it makes predictions that contradict quantum mechanics. To avoid confusion, we think that the discussion of Bohmian mechanics with regard to the above issue needs further clarification.

We have slightly expanded our discussion of how the experiment can be viewed within Bohmian mechanics. As explained in Section II.C and in the Methods section, Bohmian mechanics either violates (Q) or (C), depending on whether the agents take an outside perspective (modelling themselves as physical systems) or not.

It does not seem to be clearly explained within the manuscript the reason why W' and W carry out their measurements at two different times. Does the time order matter?

According to standard quantum theory, the time order indeed does not matter, i.e., the entries of Table II do not depend on it. However, we decided to fix a particular timing because the time order can make a difference in certain interpretations. An example is Bohmian mechanics. While a detailed discussion of this would go beyond the scope of this paper, we have added a small remark in the part that describes Bohmian mechanics.

Is the assumption (S) necessary? Even in a many-worlds interpretation, it seems clear to us that each observer in each world only sees one outcome.

We also think that, according to the common understanding, each observer sees *in each world* only one outcome. However, assumption (S) should be understood as “each observer sees only one outcome,” i.e., without “in each world.” Whether this makes a difference depends on the notion of

“world” that one uses. Since there exist, even among “many-worlders”, many views on how to define a “world”, we decided to leave a detailed analysis of the implications of our thought experiment for the various variants of many-world interpretations to future work.

Reviewer #3

p2, l56: naturally – > natural

Done.

p3, l69: The phase in the superposition of $|heads\rangle$ and $|tails\rangle$ are specified “for completeness”, which suggests that it doesn’t matter what it is. But I think the statement on p6, l113 wouldn’t be true with a different phase.

Thanks for spotting this. The phase indeed matters.

p3, eq6: Hasn’t F’s measurement happened by time $n:10$, so that there would be entanglement with F? This looks more like the state at, say, $n:05$.

The idea was that the times indicated in the description of the experimental protocol are the starting times of each step (see also Fig. 4). So, at time $n:10$, F would just start the measurement. We now clarified this in the text.

p10: I’m sceptical of the claim that the experiment would be feasible if quantum computers played the role of the friends. Either the term “quantum computer” includes the final measurements used to obtain the result of the computation, in which case the W measurements would be practically impossible as usual, or the term only refers to the unitary circuit, in which case I don’t see any reason to think the computer obtains outcomes.

This is indeed debatable, and it touches on the general question whether agents can be replaced by computers (and what a measurement means in this context). So, this last paragraph should be understood as an outlook rather than as a claim. In view of this as well as the corresponding comment by Reviewer #1, we have rephrased this part.

p13: It would be nice if there were some details of the calculations in the Bohmian and Consistent Histories cases, space permitting.

We have only slightly expanded this part. Our aim was to give an overview on what the different interpretations of quantum mechanics have to say about the experiment. However, to keep a balance between the various interpretations, we decided not to go into a too detailed discussion for any of them.

REVIEWERS' COMMENTS:

Reviewer #1 (Remarks to the Author):

The points raised in my previous review have been satisfactorily addressed by the authors. Thus, I recommend publication of this version of the manuscript.

Comments to the Authors:

The revised version of the manuscript is much improved. However, the presentation still lacks clarity. Bub's recent paper arXiv:1804.03267v1 cuts through the complicated explanations in the present manuscript by simply writing down the quantum-mechanical states. The present manuscript confuses the reader also by mixing the technical presentation of the quantum-mechanical states with the presentation of Assumptions C, Q and S, which have interpretational and philosophical significance.

We also feel that an issue of principle needs clarification. The authors wrote, "We do not think that the two measurements, the one by agent W' and the one by agent W , are incompatible. The reason is that the measurement of W' (\bar{W} in our notation) acts on lab L' (\bar{L} in our notation), whereas the measurement of W acts on lab L (see the description of the experimental protocol on page 3). These two systems, L' and L , are separate from each other (they are even at different spacial locations). We hope that these aspects are clarified by the new Figure 4."

We apologize that our criticism was unclear. We see an incompatibility not between the measurements performed by W and \bar{W} , but between the measurements leading to the statements P1 and P4. (See our previous comment and note that P1 and P4 correspond respectively to $s_Q^{\bar{W}}$ and S_Q^W of the manuscript.) In the authors' calculations, statement P1 is obtained when F uses measurement basis $|\uparrow\rangle, |\downarrow\rangle$ to measure the spin of the electron, while statement P4 is obtained when W measures L (which contains F) using the "Bell basis". These two measurements are incompatible (i.e. their corresponding observables do not commute), so that it is impossible to assign simultaneous/joint truth values to P1 and P4, as is (implicitly) allowed by their Assumption C. This is what we meant when we wrote (in the previous comments) that P1 and P4 are obtained in two incompatible contexts (thus involving counterfactual inference).

The authors might want to see Brukner's recent paper, arXiv:1804.00749v1: "A no-go theorem for observer-independent facts", wherein he derived a result similar to theirs. Brukner makes explicit the incompatibility between the spin measurement of Wigner's friend (F in the author's notation) in the basis $|\uparrow\rangle, |\downarrow\rangle$, and the measurement in the Bell basis performed by Wigner (W in the author's notation) on a system containing F and the spin. This incompatibility forbids the definition of a joint probability over the whole space of measurement outcomes of Wigner and his friend, leading to the violation of Bell's inequalities in Brukner's scheme. We agreed with Brukner that the authors' Assumption C (consistency) has a similar significance as Brukner's assumption of "observer-independent facts", allowing the combination/comparison of the measurement outcomes obtained by different agents within a single coherent story. This is done by jointly assigning their truth values via transitivity of implication, or defining joint probability over the space of all the measurements by the different agents. We also think that, as suggested by the other Referee, it might be better to avoid the word "consistency" in the authors' Assumption C, for it might give a misleading impression that a theory that violates Assumption C is an inconsistent theory. It might be worth to consider an expression such as Brukner's "observer-independent fact" a violation that forces us to see measurement outcomes as observer-dependent.

Of minor note:

There is no word "gedankenexperiment" in any language. The German word is "Gedankenexperiment" (since German nouns are capitalized). We would suggest, as alternatives, either "Gedanken experiment", "conceptual experiment" or "thought experiment".

There are two references, on p. 1 just above Eq. (1) and in the caption to Fig. 1, to "non-destructive" measurements. We do not understand what the authors mean by this term. It suggests a measurement on a state which is an eigenstate of the measured observable, but the measurement in question is *not* always on an eigenstate.

On p. 4, on lines 90-91, appears a statement that doesn't make sense to us: "In the case where $r =$ tails, and leaving out the measurements by \bar{W} and W , the situation is identical to the one considered by Wigner...". Wigner's example would require either F and W or \bar{F} and \bar{W} , not F and \bar{F} .

On p. 11, line 247 and below, the casino presumably offers \$1,000 and not \$1.000.

Reviewer #3 (Remarks to the Author):

The authors have satisfactorily addressed the minor points in my report. In my opinion the responses to the other referees are also reasonable.

In particular I think it's right to say that the theorem does not use counterfactual reasoning of the type used, for example, in Bell's theorem. However, although it's true that W and W_{bar} 's measurements are compatible, there's a sense in which W and F 's, for example, are not. But this is still different from other no-go theorems because all the measurements are actually performed.

For the reasons given in my original report, I support publication.

Response to Referee 2

We thank the referee for the additional comments on our manuscript. The following point-by-point response includes a description of the changes we have made based on the referee's remarks and suggestions. Note also that, following a request by the editor, the introduction has been restructured to comply with the journal guidelines.

The revised version of the manuscript is much improved. However, the presentation still lacks clarity. Bub's recent paper arXiv:1804.03267v1 cuts through the complicated explanations in the present manuscript by simply writing down the quantum-mechanical states.

We agree that the explicit states as described in Bub's recent paper are useful to gain an intuitive understanding of the situation. In discussions with colleagues about our work, we have however noticed that they can also be misleading. The reason is that the statements that we need for our argument (the ones of Table 3 in the revised version) follow from relations between measurement outcomes obtained at different times. For example, the first row of the table relates the value of r at time $n:01$ to the value of w at time $n:31$. Note that these two values are at no time simultaneously available. The relation between them can therefore not be read off directly from a joint quantum state at one particular time, unless additional assumptions are introduced (which we would like to avoid).

We understand that this point has not been made clear in our previous version of the manuscript. We have therefore rewritten the section on the analysis of the thought experiment and also restructured the corresponding tables to emphasise the fact that the measurement outcomes are not in general available simultaneously (see also the reply to the next to next comment).

The present manuscript confuses the reader also by mixing the technical presentation of the quantum-mechanical states with the presentation of Assumptions C, Q and S, which have interpretational and philosophical significance.

The analysis of the thought experiment strongly relies on the Assumptions C, Q, and S. For example, without assuming the validity of the quantum-mechanical Born rule, which is the essence of Assumption Q, we would not be able to derive any of the statements given in Table 3. We would therefore argue that the technical argument is not possible without stating these assumptions.

In the revised version, we have restructured the analysis section and made more explicit at what point which assumption is necessary to proceed with the technical argument.

We also feel that an issue of principle needs clarification. The authors wrote, "We do not think that the two measurements, the one by agent W' and the one by agent W , are incompatible. The reason is that the measurement of W' (\bar{W} in our notation) acts on lab L' (\bar{L} in our notation), whereas the measurement of W acts on lab L (see the description of the experimental protocol on page 3). These two systems,

L' and L , are separate from each other (they are even at different spacial locations). We hope that these aspects are clarified by the new Figure 4.”

We apologize that our criticism was unclear. We see an incompatibility not between the measurements performed by W and \bar{W} , but between the measurements leading to the statements $P1$ and $P4$. (See our previous comment and note that $P1$ and $P4$ correspond respectively to $s_Q^{\bar{W}}$ and s_Q^W of the manuscript.) In the authors’ calculations, statement $P1$ is obtained when F uses measurement basis $|\uparrow\rangle, |\downarrow\rangle$ to measure the spin of the electron, while statement $P4$ is obtained when W measures L (which contains F) using the “Bell basis”. These two measurements are incompatible (i.e. their corresponding observables do not commute), so that it is impossible to assign simultaneous/joint truth values to $P1$ and $P4$, as is (implicitly) allowed by their Assumption C. This is what we meant when we wrote (in the previous comments) that $P1$ and $P4$ are obtained in two incompatible contexts (thus involving counterfactual inference).

We thank the referee for the clarification of their earlier comment, which we indeed misunderstood. It is of course correct that the observable of the measurement leading to z does not commute with the one corresponding to the measurement of w . This also means that the values z and w cannot be defined *simultaneously* (in the sense of “at the same time”).

To explain how our argument avoids this problem, note first that the measurements take place at different times (as pointed out in the reply above). Concretely, z is measured at time $n:10$ and w is measured at time $n:30$. Note also that, in all statements we are using, we always explicitly specify the time at which a variable is supposed to have a given value. Whenever we are talking about z , the corresponding time lies (strictly) before $n:30$, whereas we only talk about what value w has at times after $n:30$. We hence never need to assume that z and w are available at the same time.

To clarify this in the manuscript, we have changed the labelling of all statements. They now have a superscript which indicates at what time a statement has been made by an agent.

The authors might want to see Brukner’s recent paper, arXiv:1804.00749v1: “A no-go theorem for observer-independent facts”, wherein he derived a result similar to theirs. Brukner makes explicit the incompatibility between the spin measurement of Wigner’s friend (F in the author’s notation) in the basis $|\uparrow\rangle, |\downarrow\rangle$, and the measurement in the Bell basis performed by Wigner (W in the author’s notation) on a system containing F and the spin. This incompatibility forbids the definition of a joint probability over the whole space of measurement outcomes of Wigner and his friend, leading to the violation of Bell’s inequalities in Brukner’s scheme. We agreed with Brukner that the authors’s Assumption C (consistency) has a similar significance as Brukner’s assumption of “observer-independent facts”, allowing the combination/comparison of the measurement outcomes obtained by different agents within a single coherent story. This is done by jointly assigning their truth values via transitivity of implication, or defining joint probability over the space of all

the measurements by the different agents. We also think that, as suggested by the other Referee, it might be better to avoid the word “consistency” in the authors’ Assumption C, for it might give a misleading impression that a theory that violates Assumption C is an inconsistent theory. It might be worth to consider an expression such as Brukner’s “observer-independent fact” a violation that forces us to see measurement outcomes as observer-dependent.

Brukner’s assumption of “observer-independent facts” is indeed related to our Assumption C. In particular, assuming “observer-independent facts” implies C. Crucially, however, the converse is not true, i.e., Assumption C does not imply “observer-independent facts”. To see this, it is sufficient to note that substituting the assumption of “observer-independent facts” in Brukner’s argument by Assumption C would not suffice to obtain his conclusion. (This is also the case for the alternative argument presented in the appendix of Brukner’s paper.) Assumption C is hence strictly weaker than Brukner’s assumption of “observer-independent facts”.

In any case, we agree with the reviewer that it is problematic to call Assumption C just “consistency”. Therefore, whenever using the term, we kept an eye on putting it in the correct context. For example, in the abstract, we are writing “The agents’ conclusions [...] are thus inconsistent”, stressing the fact that we are talking not just about consistency, but consistency between the different agents. Note also that, in the discussion section, we briefly state Brukner’s result (referring to Ref. 9, which basically contains it already).

Of minor note: There is no word “gedankenexperiment” in any language. The German word is “Gedankenexperiment” (since German nouns are capitalized). We would suggest, as alternatives, either “Gedanken experiment”, “conceptual experiment” or “thought experiment”.

We thank the reviewer for pointing this out. From the Nature Communications style guide, we understand that it should be written as *Gedankenexperiment* (in italics). But we leave it to the editor to decide here.

There are two references, on p. 1 just above Eq. (1) and in the caption to Fig. 1, to “non-destructive” measurements. We do not understand what the authors mean by this term. It suggests a measurement on a state which is an eigenstate of the measured observable, but the measurement in question is not always on an eigenstate.

Our intended meaning of “non-destructive” was that, conditioned on a particular measurement outcome, the spin should be in the state corresponding to this outcome (e.g., after obtaining measurement outcome $+1/2$, the spin is $|\uparrow\rangle$). We realise however that this use of the term may be confusing. Since it is anyway not necessary for our argument, we have omitted it in the revised version.

On p. 4, on lines 90-91, appears a statement that doesn’t make sense to us: “In the case where $r = \text{tails}$, and leaving out the measurements by \bar{W} and W , the situation is identical to the one considered by Wigner...”. Wigner’s example would require either F and W or \bar{F} and \bar{W} , not F and \bar{F} .

We meant that, if $r = \text{tails}$, then the spin is prepared as an equal superposition of $|\uparrow\rangle$ and $|\downarrow\rangle$, which is analogous to the superposition state in Wigner's original experiment. Furthermore, Wigner did not consider further measurements on the labs, which is why we wrote that the measurements by \bar{W} and W are left out. We reformulated this in the revised version and hope that it is now clearer.

On p. 11, line 247 and below, the casino presumably offers \$1,000 and not \$1.000.

Thanks for making us aware of this error, which we corrected. (We also changed the currency to Euros.)